# Cholecalciferol or Calcifediol in the Management of Vitamin D Deficiency

**DOI:** 10.3390/nu12061617

**Published:** 2020-05-31

**Authors:** Manuel Sosa Henríquez, M. Jesús Gómez de Tejada Romero

**Affiliations:** 1Investigation Group on Osteoporosis and Mineral Metabolism, University of Las Palmas de Gran Canaria, 35001 Las Palmas, Spain; 2Bone Metabolic Unit-Hospital University Insular, 35016 Gran Canaria, Spain; 3Department of Medicine, University of Seville, 41004 Seville, Spain; mjgtr@us.es

**Keywords:** vitamin D, cholecalciferol, calcifediol, hypovitaminosis D

## Abstract

Vitamin D deficiency is a global health problem due to its high prevalence and its negative consequences on musculoskeletal and extra-skeletal health. In our comparative review of the two exogenous vitamin D supplementation options most used in our care setting, we found that cholecalciferol has more scientific evidence with positive results than calcifediol in musculoskeletal diseases and that it is the form of vitamin D of choice in the most accepted and internationally recognized clinical guidelines on the management of osteoporosis. Cholecalciferol, unlike calcifediol, guarantees an exact dosage in IU (International Units) of vitamin D and has pharmacokinetic properties that allow either daily or even weekly, fortnightly, or monthly administration in its equivalent doses, which can facilitate adherence to treatment. Regardless of the pattern of administration, cholecalciferol may be more likely to achieve serum levels of 25(OH)D (25-hydroxy-vitamin D) of 30–50 ng/mL, an interval considered optimal for maximum benefit at the lowest risk. In summary, the form of vitamin D of choice for exogenous supplementation should be cholecalciferol, with calcifediol reserved for patients with liver failure or severe intestinal malabsorption syndromes.

## 1. Introduction: Vitamin D Deficiency Globally and in Our Care Environment

At present, vitamin D deficiency, in some cases even severe, is significantly prevalent worldwide. From the enormous amount of scientific literature that supports the above, we would highlight the extensive review of epidemiological studies on vitamin D status conducted in Europe, South America, North America, Asia and Oceania published by Hilger J et al. in 2014, which estimated that 88.1% of the world’s population would have levels of 25(OH)D (25-hydroxy-vitamin D, also called calcidiol or calcifediol, a metabolite determined in plasma as a biomarker of vitamin D status) below 30 nanograms/milliliter (ng/mL) [1] (minimum level of 25-hydroxy-vitamin D considered optimal with a certain degree of consensus) [2]. What is most relevant from a public health point of view, is that according to this review, 37% of the world’s population would be below 20 ng/mL (cut-off vitamin D level for deficiency, according to the most widely accepted criterion [2]), and even up to 6.7% of individuals would have 25(OH)D levels below 10 ng/mL [1], a very severe vitamin D deficiency that clearly puts the individual’s health at risk, both at the level of alterations in the musculoskeletal metabolism, and also in relation with the increasingly numerous extra-skeletal benefits that are being discovered in relation to vitamin D [2], and from which patients with such low vitamin D levels would be deprived.

This state of globally endemic hypovitaminosis D is no different in our clinical care setting, Spain, a territory where despite the belief that extensive sun exposure minimizes risk, we have vast epidemiological evidence that strictly contradicts the foregoing, highlighting a relevant prevalence of vitamin D deficiency [3].

In this regard, it is worth highlighting the data published by González-Molero I et al. in 2011, in which, in a population of 1262 healthy individuals from Asturias (in northern Spain) and Andalusia (in southern Spain) in a very wide age range of between 18 and 83 years, the average level of 25(OH)D recorded was close to the threshold of vitamin D deficiency and was far from the level considered optimal of 30 ng/mL: 22.46 ng/mL [4]. Notwithstanding the foregoing, it is true that the degree of hypovitaminosis D has been shown to be much more pronounced in at-risk populations such as post-menopausal women, in whom a very low mean level of 13.4 ng/mL was recorded in an epidemiological study carried out on 171 women between 47 and 66 years old [5]; or in institutionalized elderly people, among whom even lower levels were observed (mean): 10.2 ng/mL) with a massive prevalence of 87% of hypovitaminosis D, in about a hundred subjects admitted to a nursing home at around 78 years old [6].

Paradoxically, moreover, population groups apparently unrelated to hypovitaminosis D, such as healthy university medical students living in an area with a large solar exposure such as the Canary Islands, are not exempt from this situation. In a cross-sectional epidemiological study conducted on this population (*n* = 103), it was observed that the mean level of 25(OH)D did not reach 30 ng/mL (27.9 ng/mL); and a relevant issue factor was highlighted as even more relevant by the investigators: up to 32.6% of the individuals had lower levels than 20 ng/mL; in other words, they clearly had vitamin D deficiency [7].

Of the many reasons that could explain the above we would highlight, first of all, the very low dietary vitamin D intake obtained through food due to the fact that we know that there is a limited amount of foods that contain a sufficiently high quantity of vitamin D [8]. In fact, according to the Spanish data from the ANIBES study (Anthropometry, Intake and Energy Balance in Spain), carried out with nutritional data of just over 2000 individuals, the average vitamin D intake in Spain can be estimated at around 4.4 µg daily (equivalent to 176 International Units -IU- of vitamin D). The ANIBES investigators estimated that 93% of the Spanish population had an intake below 80% of the amount of vitamin D recommended by the health authorities both in Spain and Europe [9].

In addition, and perhaps more importantly than nutritional factors, we should highlight the factors related to the endogenous production of vitamin D3 (a molecule also known as cholecalciferol) in the skin mediated by sun exposure thanks to ultraviolet B (UVB) rays. However, in addition to the well-known fact that the majority of vitamin D reserves at the physiological level come from this endogenous cutaneous production, there is also the well-studied phenomenon of the mutagenic effect on DNA that UVB rays can produce at the skin, with the consequent risk of excessive sun exposure resulting in photo-aging and, eventually, even skin cancer including melanoma [10,11].

Sun exposure through sufficient UVB irradiation can, therefore, be harmful because of its potential risk of skin cancer (including melanoma) and also it is especially difficult to be implemented in real life since this should be done in the middle of the day (around 12 p.m.–2 p.m.), which is when the incidence of UVB rays is most perpendicular and thus most efficient in terms of the ability to endogenously produce vitamin D3 in the skin [12]. In practice, what really happens, even in countries in the northern hemisphere with high sun exposure, is that in almost half of the year (months with little sun during autumn and winter) the incidence of UVB rays through sun exposure is clearly insufficient to provide adequate endogenous production; and in the other half of the year (sunny months of spring and summer), although there may be sufficient solar irradiation, the reality is that because of the mentioned precautions mentioned from the risk of skin cancer, people sunbathe for less time than would be necessary and at times other than the central hours of the day, when UVB rays have a maximum efficiency of vitamin D production but, at the same time, also have a higher potential mutagenic risk. Moreover, and logically following the widespread recommendations of dermatological societies to minimize skin damage, people are more likely to sunbathe after applying sun creams with very high protective factors, which dramatically reduces the capacity of UVB rays to reach the deep layers of the epidermis and produce vitamin D3 physiologically from 7-dehydrocholesterol as vitamin D3 precursor [2,8,12].

Additionally, there are genetic factors in relation to skin color. Darker-skinned people (a fairly increasing native population in our clinical care environment to which more and more darker-skinned migratory populations from South America or Africa are added) have a lower vitamin D3 cutaneous production efficiency than lighter-skinned people. This group of people needs more sun exposure, compared to lighter-skinned people, in order to physiologically obtain sufficient amounts of vitamin D3. So, to have a darker skin color is an added risk factor for hypovitaminosis D [12].

In conclusion, due to all the genetic and environmental factors previously exposed, there is a clear need for exogenous therapeutic vitamin D supplementation in an increasing proportion of patients, and from a medical point of view, it is relevant to evaluate the different available therapeutic options in order to select the most appropriate one in accordance with the highest level of achieved scientific evidence.

## 2. Treatment of Hypovitaminosis D; Two options: Cholecalciferol and Calcifediol

Cholecalciferol (vitamin D3) is like ergocalciferol (vitamin D2), which we know as native vitamin D [13]. In this context, the word “native” relates to the fact that we can obtain vitamin D in the form of cholecalciferol by ingesting foods mostly of animal origin containing cholecalciferol (mainly oily or bluefish, egg yolk, fungi or meat, among others) or vegetable foods containing ergocalciferol [8,14,15]. The problem, as noted above, is that there are small amounts of vitamin D in the limited list of foods that contain it. In fact, these amounts of vitamin D obtained from a regular diet fall far short of the minimum daily intakes recommended by most scientific societies and regulatory bodies [8].

Therefore, we can affirm that cholecalciferol (vitamin D3) is the predominant form of vitamin D in nature since it is what we mammals produce in our skin which is, apart from the limited vitamin D amounts from food, the physiological and endogenous production process mediated by UVB-type solar radiation for which 7-dehydrocholesterol is converted into pre-vitamin D3, which undergoes thermal isomerization into cholecalciferol (or vitamin D3) [16].

On the other hand, the other alternative for exogenous supplementation, calcifediol, is the result of the hydroxylation of cholecalciferol in its carbon 25 position, forming 25-hydroxy-vitamin D3, a molecule known as calcifediol or calcidiol.

At the physiological level, this conversion is mediated by the 25-hydroxylase enzyme in its liver site, but also in many other body tissues. This enzyme intervenes in the introduction of a hydroxyl group in the carbon 25 position of cholecalciferol synthesized at the cutaneous level (majority), cholecalciferol from the diet (generally scarce), and also ergocalciferol obtained through the diet (generally very scarce). Therefore, two metabolites of vitamin D can be formed at the hepatic, and also extra-hepatic levels: 25-hydroxy-cholecalciferol (or 25-hydroxy-vitamin D3) or 25-hydroxy-ergocalciferol (or 25-hydroxy-vitamin D2). The sum of the two is what we generically know as 25-hydroxy-vitamin D (25(OH)D), being this the metabolite used as a measure of vitamin D status [8,16].

It is well known that 25(OH)D is the immediate precursor of an active form of vitamin D, which is formed in the kidney and also in numerous other body tissues from a second hydroxylation (this time in the carbon 1 position) mediated by 1-hydroxylase which forms 1,25-dihydroxy-vitamin D (or calcitriol), a molecule known as active vitamin D or hormone D. Calcitriol is a powerful calcitropic hormone that acts as a steroid, that is, through its binding with great affinity to the vitamin D receptor (VDR), a member of the nuclear receptor superfamily. On the basis of this union, active vitamin D directly produces the vast vitamin D beneficial effects on the health, both at the level of mineral-bone metabolism and also at the extra-skeletal level. Calcitriol also binds to caveolar receptors in cell walls, activating rapid and relevant non-genomic effects [8,16].

Ergocalciferol (the other native form of vitamin D, apart from vitamin D3, which we call vitamin D2) is the form of vitamin D that can be obtained primarily through some vegetal foods [8] and is used extensively in the United States as the form of exogenous vitamin D supplementation; however, its therapeutic use in Europe (including Spain) is virtually non-existent.

Therefore, at present, the reality is that in our clinical setting we have only two relevant therapeutic options to evaluate in terms of efficacy, efficiency, and safety when making the best medical decision concerning which of the two drugs we should use for the exogenous supplementation with vitamin D in the increasing number of patients for whom it is therapeutically indicated, and these two options are cholecalciferol and calcifediol.

Next, we will mainly evaluate both the efficacy and safety of these two therapeutic options.

## 3. Clinical Efficacy of Cholecalciferol and Calcifediol: What Do Scientific Evidence and Clinical Guidelines Indicate to Us?

### 3.1. Musculoskeletal Effects

It is well established that vitamin D has a very important role in the balance of mineral-bone metabolism. Basically, the maintenance of adequate and physiological levels of 25(OH)D is essential for proper calcium and phosphorus homoeostasis to occur through the maintenance of physiological levels of parathyroid hormone (PTH). Vitamin D counteracts the eventual excess of PTH activity, which is related to increased bone resorption and also plays key roles in osteoblastogenesis and osteoblast maturation and subsequent bone mineralization. Vitamin D, therefore, has a critical effect on bone mineralization, adequately maintains bone mineral density at both vertebral and non-vertebral levels (especially in the hip), and therefore plays a fundamental role in the prevention and treatment of osteoporosis, this role is mainly demonstrated in the risk prevention of osteoporotic fractures [17].

In relation to the translation between the theoretical biological effect of vitamin D supplementation and the prevention of osteoporotic fractures, we have carefully examined the scientific literature for publications of prospective, randomized, placebo-controlled clinical trials (RCTs), or systematic reviews of the literature and meta-analyses of RCTs performed with appropriate methodology thereof, in high-impact journals, discerning whether the prospective exogenous supplementation was performed with cholecalciferol or calcifediol.

By using this methodology, we have found several publications showing positive results that demonstrate that cholecalciferol supplementation (with or without calcium) can reduce the risk of osteoporotic fracture in a statistically significant and, at the time, clinically relevant manner. In chronological order of publication: the meta-analysis of Chapuy MV et al., published as early as 1992 in the New England Journal of Medicine, which demonstrated a statistically significant reduction in the risk of non-vertebral fractures, including hip fracture, with cholecalciferol and calcium [18]; the meta-analysis published by the Dawson-Hugues B et al. group in 1997, demonstrating a significant reduction in the risk of non-vertebral fractures of cholecalciferol and calcium [19]; the meta-analysis of the Bischoff-Ferrari HA et al. with demonstration of significant reduction in the risk of non-vertebral fracture, including hip fracture, with doses higher than 700 IU/day of cholecalciferol [20]; these results corroborated in a subsequent meta-analysis of the same group with cholecalciferol supplementation doses even higher than 400 IU/day [21]; and, finally, to mention a publication of great relevance, data from more than 30,000 women who participated in the large RCT WHI (Women’s Health Initiative), published by Prentice RL et al. in 2013, in which a 35% significant reduction in the relative risk of hip fracture was observed in patients randomized to treatment with cholecalciferol and calcium, with this risk reduction increased to 76% in those patients who were considered sufficiently adherent to exogenous supplementation with cholecalciferol and calcium [22].

On the contrary, and as far as we know, this scientific evidence has not been demonstrated and published with calcifediol supplementation (whether accompanied by calcium or not). In fact, the most relevant RCT published with calcifediol in high impact journals regarding the possible reduction of fracture risk was not entirely positive. In 2000, Peacock M et al. failed to demonstrate that calcifediol significantly reduced the risk of fractures relative to placebo in a RCT with three treatment arms (calcifediol, calcium, or placebo). In this RCT, 377 patients aged 60 years and over were randomized to receive daily doses of calcium 750 mg (*n* = 124), calcifediol 15 µg (*n* = 124), or placebo (*n* = 129) during a long-term follow-up of 17 months. The authors concluded that the supplementation with calcifediol was not superior to placebo in terms of non-vertebral or vertebral fracture risk reduction (0.680) [23]. Anyway, it is important to note here that in this Peacock trial, as in almost all of the RCTs searching for the impact of vitamin D supplementation on outcomes, there is a lack of precise presentation of baseline 25(OH)D values in normal, insufficient or deficient ranges.

In addition to reducing the risk of fractures, the amount of scientific evidence published in high impact journals regarding the relationship of cholecalciferol supplementation with relevant improvements in other significant end-points for musculoskeletal functions, such as reduction of the risk of falls or improvement of muscle function, is also remarkable. Again, there are numerous published RCTs (or meta-analyses of RCTs) demonstrating that prospective supplementation with cholecalciferol can significantly reduce the risk of falls [24,25,26,27]. However, when we systematically reviewed the published evidence with calcifediol in this regard, we only found the publication of Bischoff-Ferrari HA et al. in 2016, in which it was observed that when calcifediol was added to cholecalciferol, a paradoxical and negative effect was observed, with a statistically significant increase in the incidence of falls compared to cholecalciferol alone at a dose of 24,000 IU/month, in a three-arm RCT (cholecalciferol 24,000 IU/month, cholecalciferol 60,000 IU/month, and calcifediol 300 µg/month plus cholecalciferol 24,000 IU/month) performed on about 200 post-menopausal women [28]. This paradoxical phenomenon might be explained because of the enhanced 24-hydroxylase expression effect due to the calcifediol addition on top of cholecalciferol, since 24-hydroxylase is an enzyme responsible for calcitriol and 25(OH)D catabolism, leading to the physiological mechanism to avoid the hypercalcemia associated with hypervitaminosis D [2]. Moreover, most available assays to measure 25(OH)D have cross-reactivity with 24.25-dihydroxyvitamin D, which is a product of 25(OH)D catabolism. Therefore, the decreased biological action observed in the Bischoff-Ferrari HA et al. 2016 trial, could have resulted from an increase in 25(OH)D degradation, as well as from an overestimation of 25(OH)D levels due to the cross-reactivity of the assay.

As for the improvement of muscle function, some studies have also shown that cholecalciferol can improve it [27], while we have not found any relevant RCTs or meta-analysis with calcifediol on this matter.

The causes of this disparity in clinical results obtained between the two molecules could go beyond the mere difference in availability of cholecalciferol and calcifediol to conduct clinical research worldwide and really respond to relevant pharmacokinetic and pharmacodynamic differences between the two substances assessed as prescription drugs. Although these aspects will be further developed in the following section of this review, there is no doubt about the fact that it is critical to personalize the dose of cholecalciferol (40 IU per µg as we will see below in this review) according to baseline 25(OH)D levels and, on the other hand, concerning calcifediol, the intestinal vitamin D receptor could be exposed to supraphysiological doses that could markedly stimulate calcium and phosphorus absorption, among other differential effects of the two molecules.

Moreover, it is worth mentioning that almost all the pivotal clinical trials performed to demonstrate the efficacy and safety of the majority of anti-osteoporotic drugs currently available for osteoporosis clinical treatment (whether there are bisphosphonates, PTH analogs or RANK -Receptor Activator of Nuclear factor Kappa B- ligand inhibitors) have been conducted by supplementing patients with cholecalciferol, not calcifediol, as a vitamin D form. Therefore, the scientific evidence for all of these essential drugs for the clinical management of osteoporosis was obtained by associating them with cholecalciferol and, consequently, this should be the form of vitamin D to be used in combination with whatever anti-osteoporotic drug is chosen in order to optimize efficiency in terms of maximum protection against osteoporotic fractures [29,30].

Some of the above reasoning is probably part of the rationale which justifies the fact that the majority of relevant international scientific societies specialized in the clinical management of osteoporosis, both in Spain and also internationally, recommend cholecalciferol as the form of vitamin D of choice for the prevention and treatment of vitamin D deficiency (see Table 1).

### 3.2. Extra Musculoskeletal Effects

There is a large number of publications on epidemiological studies which have established robust relationships between vitamin D deficiency and the development or aggravation of numerous diseases of the skin, respiratory system, endocrine system, renal system, cardiovascular system, immune system, psychiatric diseases, neurodegenerative diseases, etc. Not only this, there are already numerous publications of RCTs or meta-analyses of very rigorously conducted RCTs in which it is observed that prospective vitamin D supplementation, especially in deficient patients, can provide a clinically beneficial effect in some of these diseases.

We have rigorously reviewed the RCTs or meta-analyses of RCTs available in this regard and, again, the majority of positive clinical results have been demonstrated with cholecalciferol as the vitamin D form.

In the field of dermatological diseases, we have found a meta-analysis of four RCTs (3 RCTs with cholecalciferol and 1 with ergocalciferol) in atopic dermatitis that has shown statistically significant improvements in disease severity according to validated and widely used scales such as SCORAD (Scoring Atopic Dermatitis) and EASI (Eczema Area and Severity Index) [35]; and recently a new RCT has been published confirming this potential role of cholecalciferol in atopic dermatitis according to SCORAD scale improvements [36].

In terms of respiratory diseases, there is wide evidence of benefit from cholecalciferol supplementation in chronic obstructive pulmonary disease (COPD), with two RCTs demonstrating statistically significant benefits in favor of cholecalciferol in several relevant clinical end-points, such as reduced rate of exacerbations and improvement of FEV1 (forced expiratory volume in the first second) [37,38], as well as another RCT showing improvements in inspiratory muscle strength and maximum oxygen consumption [39]. Also noteworthy in this clinical context is the ViDiCO RCT of Martineau AR et al., performed on 240 COPD patients and published in Lancet Respiratory Medicine in 2015, in which it was observed that cholecalciferol supplementation produced a statistically significant reduction in the risk of moderate or severe exacerbation in COPD patients with baseline levels of 25(OH)D below 20 ng/mL at the start of the clinical trial [40]. It is also relevant that in asthma, as the other most prevalent respiratory pathology, a systematic review of the literature and Cochrane meta-analysis was also performed by the Martineau AT et al. group, and that after systematically reviewing the literature and selecting nine RCTs of prospective cholecalciferol supplementation, they concluded that such supplementation statistically significantly reduced the rate of exacerbations of asthma requiring administration of systemic corticosteroids or resulting in emergency consultation or hospitalization, parameters of clear clinical relevance in the context of asthma management [41].

Additionally, in the context of endocrine system diseases, there are positive RCTs with cholecalciferol in several of the most prevalent endocrine diseases.

In type 2 diabetes, we have found RCTs in which cholecalciferol has been shown to produce clinical benefits in terms of statistically significant reductions in several relevant clinical end-points, including: systolic blood pressure and B-type natriuretic peptide levels [42]; in combination with calcium, significant reductions in serum insulin, glycosylated hemoglobin HbA1c, LDL (Low Density Lipoprotein) cholesterol, HDL (High Density Lipoprotein)/total cholesterol, and significant increases in QUICKI (Quantitative Insulin Sensitivity Check Index), HOMA (Homeostasis Model Assessment)-B (of pancreatic beta function) and serum HDL cholesterol [43]; reduction in serum triglyceride levels [44]; decrease in neuropathic symptoms according to the NSS (Neuropathy Symptom Score) scale in patients with diabetic neuropathy [45]; and, finally, in diabetic patients with coronary heart disease, and in combination with probiotics, cholecalciferol supplementation achieved improvements in depression and anxiety according to the BDI (Beck Depression Inventory) and BAI (Beck Anxiety Inventory) scales, as well as beneficial effects at the level of hs-CRP (highly sensitive C-reactive protein), plasma nitric oxide and total plasma antioxidant capacity [46].

In metabolic syndrome, one RCT has been published in which cholecalciferol supplementation in children was associated with statistically significant decreases in insulin and serum triglycerides, as well as HOMA-IR (insulin resistance HOMA) and continuous metabolic syndrome value [47]; and another RCT showed a decrease in serum triglycerides in adults [48]. A meta-analysis of 14 RCTs (12 of which were supplemented with cholecalciferol and only 2 with ergocalciferol) has recently been published concluding that vitamin D supplementation may reduce von Willebrand factor in patients with metabolic syndrome and its related disorders [49].

There is also published scientific evidence on the clinical context of morbid obesity and associated bariatric surgery showing the clinical benefit of cholecalciferol supplementation in these patients: in one RCT it was observed that cholecalciferol supplementation produced higher levels of 25(OH)D leading to a significantly lower rate of patients who developed secondary hyperparathyroidism [50].

Finally, regarding the spectrum of endocrine diseases, in primary hyperparathyroidism, there is a very interesting investigation from a Danish group, in which it was observed that prospective supplementation with cholecalciferol significantly reduced serum PTH levels, β-CrossLaps (β-CTX), and even increased the lumbar vertebral bone mineral density, in comparison with the placebo [51].

In terms of cardiovascular function, a body of strong epidemiological evidence has already been translated into clinical trials of cholecalciferol supplementation, and there is a RCT that has demonstrated that supplementation with monthly cholecalciferol at high doses over a prolonged period of one year in patients with baseline levels of 25(OH)D below 20 ng/mL, consistently reduced some blood pressure-related parameters, including systolic blood pressure, in patients with hypertension [52].

Concerning psychiatric disorders, we have strong evidence, especially in depression, with not less than ten published positive RCTs, some of them with very promising results [53,54]. A systematic review of the literature and meta-analysis of five RCTs presented at the 2018 American Congress of Psychiatry, concluded that vitamin D supplementation (cholecalciferol or ergocalciferol) could improve depressive symptoms according to validated and widely implemented psychiatric scales [55].

In the field of neurodegenerative diseases, several interesting published studies are also available. On Parkinson’s disease, for example, there is a RCT from a Japanese group in which cholecalciferol was supplemented to patients with predominantly early Parkinson’s disease resulting in a slowing of disease progression observed in those patients who had the vitamin D receptor genotypes FokI TT and FokI CT, in comparison with placebo patients [56]. Recently, a clinical trial of cholecalciferol supplementation in patients with mild cognitive impairment has been published with statistically significant improvements in some items of cognitive function for cholecalciferol patients, compared to placebo [57].

In any case, concerning neurological diseases, undoubtedly where there is more evidence showing the relationship between low levels of 25(OH)D and risk of disease onset, severity or worse prognosis is in multiple sclerosis. We have reviewed and identified several large prospective supplementation cholecalciferol RCTs currently on-going with the aim of confirming the epidemiological data already published [58], and there are some already performed and published RCTs such as the one by the Finnish Soilu-Hänninen M et al. group, which observed a clinically significant benefit of cholecalciferol supplementation in terms of reduction of targeted disease activity by reducing T1-enhancing MRI—magnetic resonance imaging—lesions in patients with multiple sclerosis on immunomodulatory treatment with interferon β-1b [59].

It is noteworthy to state that there have also been RCTs with negative results with cholecalciferol, although the majority of them performed in populations without basal low 25(OH)D levels and, generally speaking, the available epidemiological or clinical evidence does not disprove the hypothesis that cholecalciferol supplementation might improve clinical outcomes of some diseases by significantly rising 25(OH)D serum levels, especially when the patients have basal vitamin D deficiency status.

Unlike cholecalciferol, no RCT publications of patients with vitamin D deficiency treated with calcifediol have been identified with positive results for any clinically relevant end-points in atopic dermatitis, COPD, asthma, type 2 diabetes, metabolic syndrome, morbid obesity and associated bariatric surgery, primary hyperparathyroidism, hypertension, depression, Parkinson’s disease, mild cognitive impairment, or multiple sclerosis.

## 4. Cholecalciferol or Calcifediol: Dose Accuracy in IU, Pharmacological Differences, Aspects Related to Efficacy/Safety Balance, and Personalisation of Vitamin D Treatment

### 4.1. Dosage in IU with Cholecalciferol or Calcifediol

A relevant issue that has not received sufficient attention to date is the degree of accuracy of the IU dosage of cholecalciferol or calcifediol.

It is well known that the exogenous vitamin D dosage recommended by practically all clinical guidelines (maybe just because it is used in this way in the vast majority of important RCTs) is given in IUs and not in molecular mass units (micrograms -µg- as for vitamin D). In this regard, it is important to note that one of the most important arguments in favor of supplementation with cholecalciferol is that, since the gold standard for the equivalence between IUs and molecular mass was established with cholecalciferol, being 1 IU of vitamin D biological potency provided only with 0.025 µg of cholecalciferol [60] (proportionally, 40 IU = 1 µg of cholecalciferol), the accuracy of the number of IUs provided cannot be guaranteed when exogenously supplementing with calcifediol [61], simply because calcifediol is not the gold standard of equivalence between the molecular mass of the type of vitamin D supplemented and the biological potency provided by the molecule in question.

In relation to this matter, it should be noted that numerous clinical trials have been conducted comparing cholecalciferol and calcifediol to elucidate a possible equivalence between both active principles in terms of IU biological potency, with the premise that, microgram to microgram, calcifediol raises serum levels of 25(OH)D more than cholecalciferol since with calcifediol, we are orally providing the same molecule that we determine in plasma [25(OH)D].

In a recently published review of this issue, the published comparative clinical trials have been analyzed. Those studies have been conducted with a wide variety of methodologies (high or low doses, administered daily or intermittently), and in different populations with a wide range of sample sizes, all of them with the main objective of calculating the relative biological potency difference of the two molecules [12].

In this regard, we have found the first study published in 1977 in the journal Lancet by the Stamp et al. British group, in which among patients with vitamin D deficiency, a difference in biological potency in the range between 6 and 12 was observed [62]. Subsequently, a US study was published on healthy young adults around 28 years old among whom a somewhat lower biological potency differential was observed, in the range between 3.5 and 8 [63]; the work of Cashman et al. in adults over 50 years old, with a difference in biological potency of 4.2–4.99 [64]; the Swiss studies of Bischoff-Ferrari HA et al. in post-menopausal women with a relative potency difference: 3,4 [65]; Jetter A et al. in women aged 50 to 70 years with a relative power difference in the range between 2.23 and 5.59 [66]; the Italian work of Rossini M et al. on post-menopausal women with vitamin D deficiency, with a very low relative potency difference of 1.66 [67]; and more recently, two clinical trials published in 2017 of young American adults around 35 years old, with a relative potency difference of 5.54 [68]; and in a study of Italian Caucasian post-menopausal women over 55 years old, with a potency difference ranging from 2.8 to 8 [69]. Finally, published in 2018, the comparative clinical trial of overweight or obese Dutch adults over 65 years old, with a relative potency difference estimated between 1.04 and 2.97 [70].

There is also a clinical study carried out in our own clinical care setting that was performed with a similar methodology to those mentioned above. This work by Navarro-Valverde C et al. in 40 post-menopausal women with osteoporosis who were treated for 12 months with 20 µg (800 IU) of cholecalciferol daily, 20 µg of calcifediol daily, 266 µg of calcifediol weekly, or 266 µg of calcifediol fortnightly, concluded that the relative difference in biological potency could be estimated in a 3 to 6 range. Furthermore, the authors inferred that the calcifediol medicinal product available in Spain with 266 µg of calcifediol corresponding to 16,000 IU (according to the Summary of Product Characteristics (SPC) authorized by the Spanish Agency of Medicines and Health Products) could, in fact, have an amount of IUs clearly underestimated by the Spanish health authority, recommending its dosage by means of µg instead of IUs, the latter probably being quite inaccurate [61]. It is important to note here that the fact that most trials do not present urinary calcium:creatinine ratios with the corresponding form of supplementation is a relevant issue since it is the best objective marker of overdosing with either vitamin D compound.

Anyway, cholecalciferol and calcifediol cannot be therapeutically compared µg to µg, nor can an accurate calculation be made of the biological potency in IUs contained in 1 µg of calcifediol, but only an estimate based on any of the comparative studies listed above. Therefore, if patients needing vitamin D supplementation are to be given an exact and controlled amount of IU, in compliance with the recommendations of the most relevant clinical guidelines for the management of vitamin D deficiency, cholecalciferol appears to be the most reasonable option.

### 4.2. 25(OH)D Levels and Safety of Calcifediol or Cholecalciferol Supplementation

There is some evidence that the administration of some calcifediol doses which are commonly used in routine clinical practice, especially when intermittent (not daily) patterns are used, may induce a rise in levels of 25(OH)D, which could be considered supraphysiological as being well above the value of 30 ng/mL, commonly accepted as the optimal cut off 25(OH)D level.

We have two documented examples of the above based on the publications of two clinical trials.

The first, the aforementioned work by Navarro-Valverde C et al., in which it was observed that the average serum level of 25(OH)D reached by post-menopausal women with a diagnosis of osteoporosis treated with 266 µg of calcifediol every two weeks (dose often used in our clinical care setting) reached 65.8 ng/mL at 6 months of treatment and continued to rise gradually to 84.2 ng/mL at 12 months [61], which is a value dangerously close to the threshold of 100 ng/mL, established by the Endocrine Society in its guideline on the management of vitamin D deficiency as a level above which patients may already have clinically relevant acute safety problems, primarily hypercalcemia, hypercalciuria and, consequently, vascular or renal calcifications and associated cardiovascular or renal disorders [8].

The second, is a work published in 2018 by Olmos JM et al. in which the same dosage pattern of two monthly administrations of 266 µg of calcifediol produced average levels of 25(OH)D of 56.2 ng/mL per year at 6 months of treatment in patients with osteoporosis supplemented with calcifediol, with a proportion of patients with levels higher than 60 ng/mL reaching 38%. Even with an administration pattern of 266 µg calcifediol/month, also assessed in this study, 6% of patients exceeded 60 ng/mL. In this sense, the authors pointed out that these elevations can be considered potentially harmful and the best way to detect possible associated risks is to frequently monitor the serum levels of 25(OH)D throughout the calcifediol administration [71].

In relation to patient safety, we believe it is important to mention the issue of the possible toxicity that could be produced by vitamin D overdoses due to eventual prescription errors (mainly by physicians), dispensation errors (mainly by pharmacists) or self-administration errors (mainly by the patients themselves), leading to patients taking high daily doses in the long term, doses that, according to a proper prescription, dispensation and/or administration should actually have been administered intermittently. In fact, we have acknowledged such errors occurring in our healthcare setting, with vitamin D intoxications in which the patients involved have reached excessively high levels of 25(OH)D, with consequent clinically relevant adverse reactions.

In relation to the above, there is even a published statement from the Spanish Pharmacovigilance System in 2013 on several cases of intoxications associated with overdoses with calcifediol [72]. One year later, a significant case of metabolic encephalopathy and renal failure secondary to calcifediol overdose was reported and published on an 81-year-old female patient diagnosed with osteoporosis who, after an osteoporotic fracture, had been mistakenly taking 266 µg of calcifediol daily for an indeterminate period, reaching levels of 25(OH)D above 100 ng/mL, according to the authors [73].

Finally, in 2019, the Spanish Agency for Medicines and Health Products issued an Information Note alerting that the Spanish Pharmacovigilance System had detected serious reports of hypercalcemia in adults associated with the administration of calcifediol used in more frequent doses than recommended in its SPC [74].

In contrast to the above, and with respect to cholecalciferol, we have scientific evidence suggesting that cholecalciferol supplementation may be less likely to produce excessively high values of 25(OH)D. In a study of healthy volunteers given a dose of 4000 IU daily of cholecalciferol, an average level of 40 ng/mL was observed at 3 months, which was maintained until the completion of the 5-month total supplementation study [75].

Corroborating the above, even a pan-European regulatory body such as the European Food Safety Agency (EFSA), in a scientific report published in 2012 on the maximum tolerable intake of vitamin D, has set this upper limit at 4000 IU daily (or equivalent doses administered weekly). According to EFSA, the intake of such doses of cholecalciferol (or ergocalciferol, as the other form of native-nutrient vitamin D) is likely to be safe (without evidence of hypercalcemia, hypercalciuria, or any other relevant adverse effect) for most adult individuals [76]. This broad upper limit of 4000 IU daily of cholecalciferol or ergocalciferol that can be administered with a large margin of safety is exactly the same as that established two years earlier in 2010 on the other side of the Atlantic by the prestigious American Institute Of Medicine (IOM) [77]. It is important to note that this upper safety limit is even extended to 10,000 IU daily, or their equivalents in intermittent doses, either weekly, fortnightly or monthly, according to the criteria of the Endocrine Society [8].

There is also a study that confirms the above considerations in which a single ultra-high dose of 100,000 IU of cholecalciferol was administered to healthy volunteers with levels of 25(OH)D reaching an average peak serum concentration of 42 ng/mL after 7 days from the bolus administration. Subsequently, the values of 25(OH)D decreased steadily, remaining in a range that could be considered physiological between 42 ng/mL and 30 ng/mL until about day 70 [78].

Therefore, we could assume that when high doses of vitamin D supplementation are necessary, either with cholecalciferol or with calcifediol, the monitoring of levels of 25(OH)D that we would have to carry out would have to be much more rigorous in the case of calcifediol, and, in the case of cholecalciferol, in doses of up to 4000 IU/day (or their weekly equivalents), it could even be reasonable not to implement such monitoring, according to the scientific criteria of EFSA and IOM.

Otherwise, vitamin D, especially in its native forms, has traditionally been considered a very safe substance. In fact, until now, the main focus had been essentially put on the negative effects on musculoskeletal health (and recently also in relation to many other aspects of extra-musculoskeletal) which can be associated with long-term maintenance of low levels of 25(OH)D, without too much attention being paid on the possible harmful effects eventually produced by long-term high levels. However, in the last decade, various epidemiological studies performed with very sound methodologies, and in some cases with very large sample sizes, have been published, showing that high levels of 25(OH)D maintained over time can also be associated with deleterious negative effects on health, apart from the known acute adverse effects associated with the disorders of calcium and phosphate homeostasis eventually caused by an excess of vitamin D activity [8,71].

Consequently, and in line with the recommendations of IOM [77,79,80], we believe that we should reasonably doubt that there may be potential health risks associated if patients reach levels of 25(OH)D above about 50 ng/mL (probably corresponding with doses above 4.000 IU per day) and that since it has been observed in properly performed studies that long-term treatment with calcifediol may lead patients to exceed these levels, cholecalciferol should be chosen as the safest choice, taking into account its well-demonstrated efficacy in the treatment of vitamin D deficiency.

### 4.3. Pharmacological Differences Between Cholecalciferol and Calcifediol and Efficacy and Safety Considerations 

Cholecalciferol and calcifediol are two chemically similar molecules that are undoubtedly related in terms of metabolism. However, considered as active drugs in authorized medicinal products that we can choose and prescribe for the exogenous supplementation in the prevention and treatment of vitamin D deficiency, these molecules are quite different, since they have different pharmacokinetic and pharmacodynamic characteristics, differences which probably can explain the different safety and efficacy that we discuss in this review.

Pharmacokinetically, the most important difference between the two drugs is the elimination half-life, in other words, the time it takes to reduce to half the amount from the initially administered dose. Although there are some reports of shorter cholecalciferol elimination half-lives [81], according to different pharmacokinetic studies, we could conclude that the elimination half-life of cholecalciferol in the whole body (around 2 months) is higher than that of calcifediol (around two weeks) [82], which is mainly due to the fact that cholecalciferol is a much more lipophilic molecule than calcifediol. This increased lipophilia allows a large proportion of the cholecalciferol produced in the skin and/or exogenously supplemented to accumulate in the adipose tissue and gradually be released as long as active vitamin D is needed [82,83]. This capacity for physiological self-regulation of the pharmaco-kinetic distribution of cholecalciferol allows that, when administered as a medicinal product, it is a very suitable drug for the administration of high intermittent doses (weekly, fortnightly or even monthly dosages), which greatly facilitate compliance and adherence to treatment [83,84].

This therapeutic equivalence, in terms of comparable elevation of 25(OH)D levels, between daily administration of cholecalciferol and their equivalent schemes in terms of IU at higher intermittent doses, has been demonstrated in several clinical trials conducted with the aim of showing this thesis. In 2008, the Ish-Shalom et al. group randomized 48 Israeli women aged about 80 years who had undergone hip fracture surgery to receive 1500 IUs daily, 10,550 IUs weekly (7 × 1500), or 45,000 IUs monthly (30 × 1500) of cholecalciferol, respectively. Baseline levels of 25(OH)D were comparable among the three groups around 15 ng/mL and prospective supplementation with cholecalciferol for 2 months produced an increase in baseline levels to 33.2 ng/mL in the 1500 IU/day group; 29.2 ng/mL in the 10,500 IU/week group; and 37.1 ng/mL in the 45,000 IU/month group. The authors did not detect any statistically significant difference between these three mean values, all the three different schemes were considered equally safe, and, therefore, authors concluded that cholecalciferol supplementation could be taken daily, weekly, or monthly in an equivalent manner [85]. Almost ten years later in 2017, a Hungarian group demonstrated the same thesis, this time in 140 adults aged around 50 years, and using dosages of 1000 IUs daily, 7000 IUs weekly (7 × 1000), or 30,000 IUs monthly (30 × 1000) cholecalciferol, respectively, for 3 months of treatment. Once again, the increases in 25(OH)D levels produced by the three supplementation dosages were statistically comparable and safe and the investigators concluded that the efficacy and safety provided by daily, weekly, or monthly cholecalciferol supplementation at equivalent doses were similar, regardless of the administration pattern [86]. Some of the most recognized and relevant clinical guidelines for the management of vitamin D deficiency have considered these data and do recognize this equivalence between daily and intermittent equivalent doses of cholecalciferol [8,31].

Therefore, probably because of its slow pharmaco-kinetic elimination caused by prolonged storage and release on demand according to physiological needs, cholecalciferol, regardless of whether the dosage given is daily or intermittent (weekly, fortnightly or monthly), can maintain for a long time, physiological 25(OH)D serum levels above 30 ng/mL but below 50 ng/mL, which as extensively discussed above, could be considered the vitamin D optimal target range [61,64,65,82,83,84,85,86,87].

Calcifediol, having a much shorter elimination half-life than cholecalciferol [82], is eliminated from the body much more quickly due to its lower lipophilia and, therefore, irrelevant storage in the adipose tissue. Consequently, calcifediol as a drug in medicinal products for vitamin D deficiency may be less suitable for administration at high intermittent doses, and perhaps only daily administration could guarantee the optimal 25(OH)D level range (30–50 mg/mL) sustained on the long term [61,65,66,67,71]. In fact, we have not found any published clinical trial which demonstrates equivalence in terms of elevation of 25(OH)D serum levels between daily doses and higher intermittent equivalent doses with calcifediol.

On the other hand, in terms of pharmacodynamics, there are also relevant differences, especially in terms of the negligible affinity of cholecalciferol for VDR and, on the contrary, the conclusive available evidence of calcifediol/25(OH)D binding and activating the VDR, although to a much lesser extent than calcitriol [12,87]. Importantly, the binding of 25(OH)D to the VDR synergizes with its activation by calcitriol, resulting in enhanced calcitriol actions. At present, we know very little about the biochemical and/or physiological consequences of calcifediol being able to bind, albeit weakly, to VDR and thus, eventually performing some functions as if active hormone D. However, we could speculate that this activity on VDR produced especially by intermittent high doses could trigger negative feedback metabolic mechanisms which would explain the negative paradoxical effects on relevant end-points of musculoskeletal function such as fractures and falls, with disappointing clinical results in the few prospective RCTs with calcifediol carried out in this clinical context [23,28].

### 4.4. Personalization of Vitamin D Deficiency Treatment

Patients with vitamin D deficiency should be treated with vitamin D in a personalized manner, as we do with any other type of drug for any other pathology, adapting drug and dosage to the personal/genetic and environmental idiosyncrasy of our concrete patients.

In this sense, although we believe that for the vast majority of patients with vitamin D deficiency the drug of choice should be cholecalciferol, we also consider that there are sub-populations of patients who could better benefit from calcifediol supplementation.

In the first place, we would highlight patients with severe hepatic insufficiency, who, due to their significantly diminished hepatic function, may have problems for the conversion from cholecalciferol/ergocalciferol to 25(OH)D mediated by the 25-hydroxylase enzyme. This possible hepatic metabolic blockage of cholecalciferol should justify the fact that these patients might benefit more from supplementation with calcifediol in order to obtain an adequate amount of active vitamin D [12,84].

Similarly, in patients with severe intestinal malabsorption syndromes, significant decreases in intestinal absorption of cholecalciferol have been observed, decreases that may not be as pronounced with calcifediol [12]; therefore, these specific patients may also benefit more from calcifediol [12,84].

### 4.5. Cost and Convenience Issues

In our clinical setting in Spain, the cost of a package of cholecalciferol, 25.000 IU 4 units, is 15.61 €; and the cost of a package of calcifediol, 0.266 µg 10 units, is 13.11 € [88]. In our opinion, this small price difference in favor of calcifediol does not change the preference for cholecalciferol, which is a drug with a better benefit/risk balance and much more positive scientific evidence available. In fact, as far as we know, no comparative study has been carried out with an adequate pharmaco-economic methodology between cholecalciferol and calcifediol, and in the event it was carried out (for example, a cost-utility analysis), we believe that probably such a study could be favorable to cholecalciferol. 

## 5. Conclusions

Based on our current knowledge, treatment of vitamin D deficiency should be aimed to maintain stable and continuous serum levels of 25(OH)D in a range of approximately 30 to 50 ng/mL, which appears to be optimal in terms of maximizing benefits and minimizing risks of vitamin D, regardless of the myriad of genetic and/or environmental factors that may influence the vitamin D status of patients.

In our opinion, and based on the available scientific evidence, cholecalciferol is the form of vitamin D that can ensure that the vast majority of patients with vitamin D deficiency are within the optimal range of efficacy and safety in a long-term period.

Therefore, based on our review of differential pharmacological characteristics and scientific evidence, cholecalciferol should be used and prescribed in the majority of vitamin D deficiency clinical settings instead of calcifediol. 

## Figures and Tables

**Table 1 nutrients-12-01617-t001:** List of relevant national and international scientific societies specialized in the clinical management of osteoporosis that recommend cholecalciferol as the vitamin D form of choice.

Scientific Society	Geographical Scope	Year of Publication	Reference
SEIOMM (Spanish Society for Bone and Mineral Metabolism Research)	Spain	2011	[17]
ES (Endocrine Society)	Global	2011	[8]
IOF (International Osteoporosis Foundation)	Global	2010	[31]
NOF (National Osteoporosis Foundation)	United States	2014	[32]
NOS (National Osteoporosis Society)	United Kingdom	2014	[33]
AACE (American Association of Clinical Endocrinologists)	United States	2016	[34]

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
