# Peer review of "Cholecalciferol or Calcifediol in the Management of Vitamin D Deficiency"

_nutrients, 2020, doi:10.3390/nu12061617_

Round 1
Reviewer 1 Report
The changes made were appropriate responses to the queries and commentss made by this reviewer.
Author Response
Thanks for the reviewer's comments.
Reviewer 2 Report
Comments to authors
The authors have appropriately addressed the most relevant concerns raised in my prior review of their work, which presents important clinical and pathophysiological bases for the implementation of safe personalized strategies to correct hypovitaminosis D. There remain two important clarifications on vitamin D metabolism and actions that need to be included in the text and, also, a few very minor changes in the text to convey the right message to the readers. Specifically:
Important modifications to implement in the text:
- Lanes 704-705: The autors present a very misleading statement in the half lifes of vitamin D and 25(OH)D in the circulation. The half life of vitamin D (both D2 and D3) upon a single bolus injection is of less than 3 days (Armas LA et al, J Clin Endocrin. Metab 89:5387, 2004), while 25(OH)D3 has a half life of 15-20 days. However as the authors correctly indicate, due to its highly lypophyllic nature, the half life of vitamin D in the whole body is of about 2 months. Please, correct.
- Lanes 764-765: Please add at the end: Importantly, the binding of 25(OH)D to the VDR synergizes with its activation by calcitriol, resulting in enhanced calcitriol actions.
Very minor corrections:
- Lane 117: Please delete the 5 after that (that5).
- Lane 138: Please replace “several target body tissues” by numerous other body tissues. These tissues are sources (rather than targets) of “local” calcitriol production for VDR activation and , therefore, decrease the risk of hipercalcemia. Please, correct.
- Lane 141: Please correct “that acts as steroid hormones” rather than “ that acts as of steroid…”
- Lane 579: Please, replace “Consequently, an…” by Consequently, and….
Author Response
Comments to authors
The authors have appropriately addressed the most relevant concerns raised in my prior review of their work, which presents important clinical and pathophysiological bases for the implementation of safe personalized strategies to correct hypovitaminosis D. There remain two important clarifications on vitamin D metabolism and actions that need to be included in the text and, also, a few very minor changes in the text to convey the right message to the readers.
Specifically:
Important modifications to implement in the text:
- Lanes 704-705: The autors present a very misleading statement in the half lifes of vitamin D and 25(OH)D in the circulation. The half life of vitamin D (both D2 and D3) upon a single bolus injection is of less than 3 days (Armas LA et al, J Clin Endocrin. Metab 89:5387, 2004), while 25(OH)D3 has a half life of 15-20 days. However as the authors correctly indicate, due to its highly lypophyllic nature, the half life of vitamin D in the whole body is of about 2 months. Please, correct.
- Lanes 764-765: Please add at the end: Importantly, the binding of 25(OH)D to the VDR synergizes with its activation by calcitriol, resulting in enhanced calcitriol actions.
All modifications are implemented.
Very minor corrections:
- Lane 117: Please delete the 5 after that (that5).
- Lane 138: Please replace “several target body tissues” by numerous other body tissues. These tissues are sources (rather than targets) of “local” calcitriol production for VDR activation and , therefore, decrease the risk of hypercalcemia. Please, correct.
- Lane 141: Please correct “that acts as steroid hormones” rather than “ that acts as of steroid…”
- Lane 579: Please, replace “Consequently, an…” by Consequently, and….
All corrections are done.
This manuscript is a resubmission of an earlier submission. The following is a list of the peer review reports and author responses from that submission.
Round 1
Reviewer 1 Report
Summary
This is a very important review of the optimal approach with vitamin D3 (colecalciferol) or its activated metabolite 25-hydroxyvitaminD, to correct the disorders associated to vitamin D deficiency, insufficiency in a daily enlarged population around the world, clearly emphasizing the two critical issues when considering the use of one of these molecules vs. the other: efficacy and safety.
Comments to authors.
The article is well written, present the most relevant clinical trials and points to the caution necessary when the choice for supplementation is calcidiol. There are several issues that require correction or clarification to avoid misleading a non expert reader. Specifically,
Lanes 124-129: correct “the hepatic 25-hydroxylase” as it were only one. Please, clarify that for many years Cyp27A1 was considered the hepatic 25-hydroxylase. It is well known now that cyp2R1 is the enzyme that contributes the most to vitamin D3 conversion to 25-hydroxyvitamin D (25D3) as mutations in Cyp2R1 but not Cyp27A1 cause severe vitamin D deficiency in humans. (Cheng JB et al PNAS (USA)2004, 101: 7711 or Thacher TD, 2015 among many others). It is also important to emphasize that: a) 25hydroxyvitamin D3 levels are used as a measure of vitamin D status because vitamin D measurements are too difficult for regular clinical laboratories and even more importantly, b) That the liver is the main, but not the only site for the conversion of vitamin D to 25D3. Lane 142: Please, indicate that calcitriol is a powerful calcitropic hormone, that acts as steroid hormones, that is through its binding to its receptor, the vitamin D receptor, a member of the nuclear receptor superfamily. Lanes 149-150: In addition to the suppression of PTH synthesis and its associated increases in bone resorption, vitamin D plays key roles on osteoblastogenesis and osteoblast maturation and mineralization potential. Please, add. Lanes 187-192. Please, clarify the dose of 25D3 used, the supplementation regimen, and most importantly, the length of the study and the age of the participants to emphasize the importance of personalizing rather than generalizing to all ages the impact of either vitamin D3 or 25D3. Please, also indicate one major failure in almost all trials searching for the impact of vitamin D supplementation on outcomes: Lack of a precise presentation of baseline vitamin D values in the normal, insufficient or defficient ranges. Please, also specify what the authors mean by “No serious” adverse effects. There were elevations in 25D3 levels above the recommended levels for safety (100 ng/ml). Lanes 198-204: The addition of calcidiol could have enhanced 24-hydroxylase expression, an enzyme responsible for calcitriol and 25D3 catabolism to avoid the hypercalcemia associated with hypervitaminosis D. In contrast with your statement below in this manuscript (lanes 581-583), there is conclusive evidence that 25D can directly actívate the VDR. Indeed, this was conclusively demonstrated in cells that lack 1-hydroxylase or that were exposed to a modified 25D3 that cannot be converted to calcitriol (Lou YR J.Steroid Biochem Mol Biol 2010; 118: 1062-1089 and Hoenderoup JC, et al Kidney Int 2004; 66: 1082-1089). Therefore, the addition of calcidiol on top of vitamin D, could have enhanced calcidiol degradation. In fact, were serum 25D3 levels measured in this trial? Also, please remember and remind the reader that most available assays to measure 25D have a 100% cross reactivity with 24,25-dihydroxyvitamin D, the product of 25D3 catabolism. Therefore, the decreased biological actions could result from an increase in 25-hydroxy-vitamin D degradation and an overestimation of 25D3 levels due to the cross reactivity of the assay. Please, clarify. Lanes 214-222: This is a very important statement and once again, it is critical to emphasize the need to personalize the dose of colecalciferol according to baseline 25D3 levels. Also, please indicate that the equivalence of 40IU= 1ug of cholecalciferol takes into consideration the rates of absorption and the rates for conversion to 25D3, while for 25D3 there is no need for conversion, only absorption rates. In the evaluation of activity, it should be taken into consideration that despite achieving similar levels in the circulation, in the case of calcidiol, the intestinal vitamin D receptor is exposed to supraphysiological doses that could markedly stimulate calcium and phosphorus absorption. This enhanced intestinal absorption may not result in increases in calcemia unless there is concommitant renal failure. The best way to evaluate a vitamin D excess is the measurement of urinary Ca/creatinine ratio. Even in aging people with mildly impaired renal function, but with abnormal bone remodeling, the excess of calcium will end up in arteries and soft tissue, increasing mortality risks. Lane 353: The statement of higher efficacy higher risks is ignoring the efficacy of vitamin D at no risk in tissues that express 25-hydroxylase as demonstrated even for tissues unrelated to mineral metabolism as the endometrium (Bergada L, et al Lab Invest 2014; 94: 608-622. Lanes 399-399 and 400-407: Most trials do not present urinary calcium:creatinine ratios with either formo f supplementation. This is the best marker of overdosing with either vitamin D metabolite. Please, clarify. Lanes426-430. Please, clarify that doses higher than 4000 IU daily will have a lower rate of conversion to 25D3 (Armas et al; J Clin Endoc Metab 2004; 89:5387 and Holick M et al, J Clin Endoc Metab 2208; 93(3)677-681). Lane 451: Control of urinary calcium/creatinine Lanes 457-461 Ref 70 is not available. Please, correct. Lane 505: A vitamin D excess impairs anti-inflammatory actions probably due to 25D5-VDR induction of 25D3 degradation, or most likely through the induction of hyperphosphatemia, a recognized pro-aging, pro-inflammatory signal. Lanes 571-578: 25D3 induces its own degradation once reaching supraphysiological levels. Lanes 581-583: Please, correct. As indicated previously, there is conclusive evidence of 25D3 binding to the VDR followed by its activation.Author Response
Summary
This is a very important review of the optimal approach with vitamin D3 (colecalciferol) or its activated metabolite 25-hydroxyvitaminD, to correct the disorders associated to vitamin D deficiency, insufficiency in a daily enlarged population around the world, clearly emphasizing the two critical issues when considering the use of one of these molecules vs. the other: efficacy and safety.
Comments to authors.
The article is well written, present the most relevant clinical trials and points to the caution necessary when the choice for supplementation is calcidiol. There are several issues that require correction or clarification to avoid misleading a non expert reader. Specifically,
Lanes 124-129: correct “the hepatic 25-hydroxylase” as it were only one. Please, clarify that for many years Cyp27A1 was considered the hepatic 25-hydroxylase. It is well known now that cyp2R1 is the enzyme that contributes the most to vitamin D3 conversion to 25-hydroxyvitamin D (25D3) as mutations in Cyp2R1 but not Cyp27A1 cause severe vitamin D deficiency in humans. (Cheng JB et al PNAS (USA)2004, 101: 7711 or Thacher TD, 2015 among many others). It is also important to emphasize that: a) 25hydroxyvitamin D3 levels are used as a measure of vitamin D status because vitamin D measurements are too difficult for regular clinical laboratories and even more importantly, b) That the liver is the main, but not the only site for the conversion of vitamin D to 25D3.
Done in lines 128-135 (see Revised manuscript).
Lane 142: Please, indicate that calcitriol is a powerful calcitropic hormone, that acts as steroid hormones, that is through its binding to its receptor, the vitamin D receptor, a member of the nuclear receptor superfamily.
Done in lines 139-141 (see Revised manuscript).
Lanes 149-150: In addition to the suppression of PTH synthesis and its associated increases in bone resorption, vitamin D plays key roles on osteoblastogenesis and osteoblast maturation and mineralization potential. Please, add.
Done in lines 177-178 (see Revised manuscript).
Lanes 187-192. Please, clarify the dose of 25D3 used, the supplementation regimen, and most importantly, the length of the study and the age of the participants to emphasize the importance of personalizing rather than generalizing to all ages the impact of either vitamin D3 or 25D3. Please, also indicate one major failure in almost all trials searching for the impact of vitamin D supplementation on outcomes: Lack of a precise presentation of baseline vitamin D values in the normal, insufficient or defficient ranges. Please, also specify what the authors mean by “No serious” adverse effects. There were elevations in 25D3 levels above the recommended levels for safety (100 ng/ml).
Partially done in lines 215-220 (see Revised manuscript). Sorry but we do not quite see how to include the comment on adverse effects in this paragraph.
Lanes 198-204: The addition of calcidiol could have enhanced 24-hydroxylase expression, an enzyme responsible for calcitriol and 25D3 catabolism to avoid the hypercalcemia associated with hypervitaminosis D. In contrast with your statement below in this manuscript (lanes 581-583), there is conclusive evidence that 25D can directly actívate the VDR. Indeed, this was conclusively demonstrated in cells that lack 1-hydroxylase or that were exposed to a modified 25D3 that cannot be converted to calcitriol (Lou YR J.Steroid Biochem Mol Biol 2010; 118: 1062-1089 and Hoenderoup JC, et al Kidney Int 2004; 66: 1082-1089). Therefore, the addition of calcidiol on top of vitamin D, could have enhanced calcidiol degradation. In fact, were serum 25D3 levels measured in this trial? Also, please remember and remind the reader that most available assays to measure 25D have a 100% cross reactivity with 24,25-dihydroxyvitamin D, the product of 25D3 catabolism. Therefore, the decreased biological actions could result from an increase in 25-hydroxy-vitamin D degradation and an overestimation of 25D3 levels due to the cross reactivity of the assay. Please, clarify.
Done in 233-241 lines (see Revised manuscript).
Lanes 214-222: This is a very important statement and once again, it is critical to emphasize the need to personalize the dose of colecalciferol according to baseline 25D3 levels. Also, please indicate that the equivalence of 40IU= 1ug of cholecalciferol takes into consideration the rates of absorption and the rates for conversion to 25D3, while for 25D3 there is no need for conversion, only absorption rates. In the evaluation of activity, it should be taken into consideration that despite achieving similar levels in the circulation, in the case of calcidiol, the intestinal vitamin D receptor is exposed to supraphysiological doses that could markedly stimulate calcium and phosphorus absorption. This enhanced intestinal absorption may not result in increases in calcemia unless there is concommitant renal failure. The best way to evaluate a vitamin D excess is the measurement of urinary Ca/creatinine ratio. Even in aging people with mildly impaired renal function, but with abnormal bone remodeling, the excess of calcium will end up in arteries and soft tissue, increasing mortality risks.
Done in lines 249-253 (see Review manuscript).
Lane 353: The statement of higher efficacy higher risks is ignoring the efficacy of vitamin D at no risk in tissues that express 25-hydroxylase as demonstrated even for tissues unrelated to mineral metabolism as the endometrium (Bergada L, et al Lab Invest 2014; 94: 608-622.
Sorry but we do not quite see how to include the comment on efficacy of vitamin D in tissues that express 25-hydroxylase in this paragraph.
Lanes 399-399 and 400-407: Most trials do not present urinary calcium:creatinine ratios with either formo f supplementation. This is the best marker of overdosing with either vitamin D metabolite. Please, clarify.
Done in lines 427-429 (see Review manuscript).
Lanes 426-430. Please, clarify that doses higher than 4000 IU daily will have a lower rate of conversion to 25D3 (Armas et al; J Clin Endoc Metab 2004; 89:5387 and Holick M et al, J Clin Endoc Metab 2208; 93(3)677-681).
Sorry but we do not quite see how to include the comment on lower rate of conversion to 25(OH)D of doses higher than 4000 IU in this paragraph.
Lane 451: Control of urinary calcium/creatinine.
Since previously included in lines 427-429, we would prefer not to mention it again in this paragraph.
Lanes 457-461 Ref 70 is not available. Please, correct.
Sorry but we are not able to fully understand this correction. Relevant reference 70 (Olmos JM, Arnaiz F, Henández JL, Olmos-Martínez JM, González-Macías J. Calcifediol mensual frente a calcifediol quincenal en el tratamiento de pacientes osteoporóticos. Estudio en la vida real. Rev Osteoporos Metab Miner. 2018;10(2):89-95. DOI: 10.4321/S1889-836X2018000200005.) is correctly included, firstly mentioned now in line 458 and also afterwards in some other paragraphs of the manuscript.
Lane 505: A vitamin D excess impairs anti-inflammatory actions probably due to 25D5-VDR induction of 25D3 degradation, or most likely through the induction of hyperphosphatemia, a recognized pro-aging, pro-inflammatory signal.
Done in lines 558-560 (see Review manuscript).
Lanes 571-578: 25D3 induces its own degradation once reaching supraphysiological levels.
Sorry but we do not quite see how to include this comment on own degradation of 25(OH)D in this paragraph.
Lanes 581-583: Please, correct. As indicated previously, there is conclusive evidence of 25D3 binding to the VDR followed by its activation.
Done in lines 641-643 (see Review manuscript).

Reviewer 2 Report
Review-Nutrients-2020-cholecalciferol of calcifediol in the management of vitamin D deficiency
This is a report aiming to determine which of the above compounds may be the most suitable for correction of vitamin D deficiency when considered as a ‘medical’ problem.
General comments.
1. This narrative review does not consider the public health measure of food fortification with vitamin D, where populations need to be protected from deficiency as cost effectively as possible and as safely as possible, though I am sure they would consider intact vitamin D, preferably cholecalciferol, to be the appropriate compound to use for this purpose. It would be useful to cover this point specifically.
2. The authors have not discussed the cost implications of ‘treating’ people with these two different compounds but should do so. No doubt pharmaceutical companies would like to sell large amounts of calcifediol, but I am not aware of any justification for using this generally, as the authors state, nor at the population level, where safety would clearly be a major problem. This compound should only be used when medically indicated, as the authors already conclude, and this point should appear as a firm conclusion in the summary/conclusions and not just in the abstract.
3. The text is reasonably clear and much of it is easy to follow though in places the meaning is confused by the way it is written, some examples being mentioned in the specific comments. However, the text is also very much too long, and the points made could almost all be made much more succinctly. Also, the length of the text does bury the nuggets of information about how giving calcifediol differs in its effects from giving cholecalciferol which is a pity.
4. There is no need to provide long descriptions of all the data available on the health benefits of adequate vitamin D status. These descriptive sections could be shortened a great deal by providing an introductory section on the overall health benefits of repletion, in brief, but with enough references for the interested reader to be able to look into such matters in more detail. I say this as this article is not primarily about what vitamin D may or may not do.
5. The main point of this report is the comparison of the two compounds discussed in terms of their potential as therapeutic agents for use in treating vitamin D deficiency [rather than in preventing it.] The main emphasis, therefore, should be on the pros and cons of the use of each of these compounds rather than on what vitamin D may do in general.
6. The arguments for using cholecalciferol rather than 25(OH)D would be more impressive if they were presented in a single section, with subsections for each aspect of those arguments.
7. The medical reasons for considering the use of calcifediol rather than of cholecalciferol needs to be presented in a clearly marked subsection. This point is mentioned in the abstract but should also be mentioned in the conclusions.
Specific comments [by line number] .These comments are not comprehensive but provide suggestions for the sort of editing that is needed.
Line 67, numbers not amounts; lines 80-81, ‘sun exposure can be negative’ is not a specific way to report the adverse effects of excessive UVB exposure; line 109, ‘native’ is not a common way to describe D3; line 112, do you mean vegetable?; line 132, ‘mostly in the kidney’ is unlikely to be correct because while activation was first discovered in the kidney, [circulating to provide hormonal effects relevant to bone health] it is now known to happen in all target tissues, though differently regulated. Thus serum 25(OH)D is the substrate for all target tissue activation and its concentration is a major factor in regulating target tissue activation. This is an important point to mention since this fact specifically increases the risk of side effects from over large intakes of 25(OH)D, while this problem is not a feature of larger intakes of intact D3 because each stage of activation is tightly regulated by feedback mechanisms; line 153, “indicate to us?” or ‘tell us’, or ‘suggest’” lines 238-327, is an example of text that would benefit greatly from being shortened.
Lines 337-384. The text here needs to discuss the general point raised in line 167 above; lines 334-384 contain text that would benefit from being shortened and more concise, which would increase the impact on readers of the comparisons being reported. This is also the case for lines 390-588.
Line 531,’ …. these molecules are quite different’; line 533, ..that we have discussed in this review; line 604 “…..benefit from treatment with calcidiol which may be better absorbed than cholecalciferol when there is fat malabsorption and, importantly, which would compensate for the reduction in hepatic 25-hydroxylation of cholecalciferol.”; line 608, is clearly better in terms of …..’; and mention when to use calcidiol, [as suggested above].
Author Response
Review-Nutrients-2020-cholecalciferol of calcifediol in the management of vitamin D deficiency
This is a report aiming to determine which of the above compounds may be the most suitable for correction of vitamin D deficiency when considered as a ‘medical’ problem.
General comments.
This narrative review does not consider the public health measure of food fortification with vitamin D, where populations need to be protected from deficiency as cost effectively as possible and as safely as possible, though I am sure they would consider intact vitamin D, preferably cholecalciferol, to be the appropriate compound to use for this purpose. It would be useful to cover this point specifically.
Indeed the scope of our review has not been food fortification with vitamin D as public measure, but efficacy and safety of vitamin D compounds habitually used in our clinic setting as vitamin D medical treatment, that is to say as medicinal products. Vitamin D food fortification would be a really interesting matter to be treated, but maybe on another paper.
The authors have not discussed the cost implications of ‘treating’ people with these two different compounds but should do so. No doubt pharmaceutical companies would like to sell large amounts of calcifediol, but I am not aware of any justification for using this generally, as the authors state, nor at the population level, where safety would clearly be a major problem. This compound should only be used when medically indicated, as the authors already conclude, and this point should appear as a firm conclusion in the summary/conclusions and not just in the abstract.
Honestly, we have not the pharmacoeconomic expertise to discuss economic implications of using calcifediol versus cholecalciferol and our comparisons and discussions have been performed from a strictly medical point of view. Moreover, we consider that our preference for cholecalciferol has been sufficiently stated throughout the manuscript.
The text is reasonably clear and much of it is easy to follow though in places the meaning is confused by the way it is written, some examples being mentioned in the specific comments. However, the text is also very much too long, and the points made could almost all be made much more succinctly. Also, the length of the text does bury the nuggets of information about how giving calcifediol differs in its effects from giving cholecalciferol which is a pity.
You may be right and the text could be shortened and organized in a different way, but the presented structure and length are the ones which have better deployed the ideas we wanted to point out with the manuscript.
There is no need to provide long descriptions of all the data available on the health benefits of adequate vitamin D status. These descriptive sections could be shortened a great deal by providing an introductory section on the overall health benefits of repletion, in brief, but with enough references for the interested reader to be able to look into such matters in more detail. I say this as this article is not primarily about what vitamin D may or may not do.
As it is well known, the potential health benefits of vitamin D are large and hardly to be presented briefly. We have tried to do it as simply as we could and focusing in our main thesis: when extra musculoskeletal health, as well as musculoskeletal, evidence is available, that evidence has been achieved with cholecalciferol, not with calcifediol.
The main point of this report is the comparison of the two compounds discussed in terms of their potential as therapeutic agents for use in treating vitamin D deficiency [rather than in preventing it.] The main emphasis, therefore, should be on the pros and cons of the use of each of these compounds rather than on what vitamin D may do in general.We humbly consider that we have been able to emphasize the pros of using vitamin D (especially cholecalciferol) and also the cons (especially calcifediol), with a major premise: vitamin D treatment, whatever compound is chosen, should be indicated only when medically required, and in a personalized manner.
The arguments for using cholecalciferol rather than 25(OH)D would be more impressive if they were presented in a single section, with subsections for each aspect of those arguments.
In our mental scheme, we have chosen to compare the drugs according to different relevant medical aspects (efficacy, safety, accuracy in IU,…) in different sections rather than in a single section because, by the way, we also wanted to cover some general aspects concerning vitamin D supplementation in those different sections.
The medical reasons for considering the use of calcifediol rather than of cholecalciferol needs to be presented in a clearly marked subsection. This point is mentioned in the abstract but should also be mentioned in the conclusions.
In our opinion, we have sufficiently presented the two niches in which we prefer the use of calcifediol rather than the use of cholecalciferol (severe hepatic insufficiency and severe intestinal malabsorption) in paragraph 4.4. (see lines 650-665).
Specific comments [by line number] .These comments are not comprehensive but provide suggestions for the sort of editing that is needed.
Line 67, numbers not amounts;
Partially done: we have considered “quantity” (see line 68 in Revised manuscript).
Lines 80-81, ‘sun exposure can be negative’ is not a specific way to report the adverse effects of excessive UVB exposure;
Done: “harmful” (see line 81 in Revised manuscript).
Line 109, ‘native’ is not a common way to describe D3;
Indeed it is not a common way to describe it, but we would like to point out that “native” could be a possible naming; that is why we mentioned it.
Line 112, do you mean vegetable?;
Yes indeed; our mistake!. Change done (see line 115 in Revised manuscript).
Line 132, ‘mostly in the kidney’ is unlikely to be correct because while activation was first discovered in the kidney, [circulating to provide hormonal effects relevant to bone health] it is now known to happen in all target tissues, though differently regulated. Thus serum 25(OH)D is the substrate for all target tissue activation and its concentration is a major factor in regulating target tissue activation. This is an important point to mention since this fact specifically increases the risk of side effects from over large intakes of 25(OH)D, while this problem is not a feature of larger intakes of intact D3 because each stage of activation is tightly regulated by feedback mechanisms;
Done (see line 137 and also section 4 in Revised manuscript).
Line 153, “indicate to us?” or ‘tell us’, or ‘suggest’”
Done: “therapeutically needed” (see line 167 in Revised manuscript).
Lines 238-327, is an example of text that would benefit greatly from being shortened.
We have considered to include this information and we humbly do not see it as redundant.
Lines 337-384. The text here needs to discuss the general point raised in line 167 above;
Sorry but we do not see the link between line 167 and content in 337-384.
Lines 334-384 contain text that would benefit from being shortened and more concise, which would increase the impact on readers of the comparisons being reported. This is also the case for lines 390-588.
We have considered to include this information and we do not consider it as redundant.
Line 531,’ …. these molecules are quite different’;
Done in Line 558 (see line 590 in Revised manuscript).
Line 533, ..that we have discussed in this review;
Partially done: “we discuss in this review” (see line 592 in Revised manuscript).
Line 604 “…..benefit from treatment with calcidiol which may be better absorbed than cholecalciferol when there is fat malabsorption and, importantly, which would compensate for the reduction in hepatic 25-hydroxylation of cholecalciferol.”;
Sorry but we are not able to understand this comment; anyway, we would like to keep on pulling apart the hepatic and the malabsorption matters in two different paragraphs.
Line 608, is clearly better in terms of …..’; and mention when to use calcidiol, [as suggested above].
As mentioned above, we consider that we have already done that in the whole 4.4 paragraph.

Round 2
Reviewer 2 Report
I feel it is a pity that the authors could not make this report more concise as readers may fail to read it properly in view of its length and it does provide answers to the important question it raises.
Many of the minor but specific suggestions made by this reviewer have been acted upon which has improved the accuracy of the report somewhat.
Some few problems in the text remain that require simple editing, as follows:-
Line 141, the authors should mention the important non genomic effects of vitamin D that are not mentioned in the text by adding a clause to this comment, e.g. 'and binds to caveolar receptors in cell walls, activating rapid non-genomic effects'
Line 167, this sentence says 'are needed'. This is not wrong but the term that is commonly used to report that a treatment is appropriate [needed] is that it is 'indicated' and, therefore, this term should be used since it has a very widely accepted meaning.
Line 436, the text has now omitted the '25' before' (OH)D' which should be replaced.
Furthermore, I still suggest that the comparative costs of 25(OH)D and intact D3 should be mentioned, even if only as a footnote;such information should be easily available from a hospital pharmacy or on line.
Author Response
Dear Sir/Madam,
We have observed all the comments and recommendations.
Kindest regards
Professor Manuel Sosa-Henríquez
